# White Matter Interstitial Neurons in the Adult Human Brain: 3% of Cortical Neurons in Quest for Recognition

**DOI:** 10.3390/cells10010190

**Published:** 2021-01-19

**Authors:** Goran Sedmak, Miloš Judaš

**Affiliations:** 1Croatian Institute for Brain Research, University of Zagreb School of Medicine, Šalata 12, 10 000 Zagreb, Croatia; mjudas@hiim.hr; 2Center of Excellence for Basic, Clinical and Translational Neuroscience, Šalata 12, 10 000 Zagreb, Croatia

**Keywords:** WMIN, subplate, human brain, stereology, epilepsy, schizophrenia, cerebral cortex

## Abstract

White matter interstitial neurons (WMIN) are a subset of cortical neurons located in the subcortical white matter. Although they were fist described over 150 years ago, they are still largely unexplored and often considered a small, functionally insignificant neuronal population. WMIN are adult remnants of neurons located in the transient fetal subplate zone (SP). Following development, some of the SP neurons undergo apoptosis, and the remaining neurons are incorporated in the adult white matter as WMIN. In the adult human brain, WMIN are quite a large population of neurons comprising at least 3% of all cortical neurons (between 600 and 1100 million neurons). They include many of the morphological neuronal types that can be found in the overlying cerebral cortex. Furthermore, the phenotypic and molecular diversity of WMIN is similar to that of the overlying cortical neurons, expressing many glutamatergic and GABAergic biomarkers. WMIN are often considered a functionally unimportant subset of neurons. However, upon closer inspection of the scientific literature, it has been shown that WMIN are integrated in the cortical circuitry and that they exhibit diverse electrophysiological properties, send and receive axons from the cortex, and have active synaptic contacts. Based on these data, we are able to enumerate some of the potential WMIN roles, such as the control of the cerebral blood flow, sleep regulation, and the control of information flow through the cerebral cortex. Also, there is a number of studies indicating the involvement of WMIN in the pathophysiology of many brain disorders such as epilepsy, schizophrenia, Alzheimer’s disease, etc. All of these data indicate that WMIN are a large population with an important function in the adult brain. Further investigation of WMIN could provide us with novel data crucial for an improved elucidation of the pathophysiology of many brain disorders. In this review, we provide an overview of the current WMIN literature, with an emphasis on studies conducted on the human brain.

## 1. Introduction

The human brain is composed of billions of molecularly, structurally, and functionally distinct neurons. In recent years, a number of large-scale transcriptomic, proteomic, and connectome studies of the cerebral cortex and its neurons were conducted to elucidate the neurobiological basis of cortical functions and our cognitive abilities. In the majority of these studies, the focus was on the neurons located within the traditionally described layers of the cerebral cortex. However, in all of these studies one significant subset of cortical neurons remains largely ignored. Upon closer inspection, it has been shown that, in addition to neurons in the six traditionally described cytoarchitectonic cortical layers, there is a large population of cortical neurons located in the subcortical white matter. These neurons, in the adult human brain, are described as white matter interstitial neurons (WMIN). While WMIN have also managed to capture a certain level of interest when it comes to scientific research, they are still largely unexplored, and their possible significance is often underestimated. For example, WMIN were recently reviewed in the “Brain Mythology” section of a renowned neuroscience journal [1].

It should be noted that WMIN were originally described over 150 years ago [2] and that they were initially proposed as a normal neuronal component of the subcortical white matter (for a detailed historical review, see [3]). However, in 1910 [4] it was suggested that WMIN may in fact represent a pathological finding in the subcortical white matter. Thus, even today, we have to deal with two seemingly conflicting conceptions of WMIN, one defining them as a normal (but minor) population of cortical neurons with undefined/unknown function, and the other as misplaced remnants of fetal migratory neurons with a potentially pathogenetic role in the adult brain.

## 2. Developmental Origin, Morphology, and Molecular Profile of WMIN

In elucidating the developmental origin of WMIN, the discovery of a novel subplate zone (SP) [5] had a crucial role (for a detailed history of that discovery, see [3]). The subplate zone represents a key transient fetal compartment responsible for normal cortical development [6,7,8]. The SP is composed of mature postmitotic neurons, afferent and efferent axons, and abundant extracellular matrix. The SP neurons exhibit both glutamatergic and GABAergic profiles. It is important to note that, besides being among the earliest generated neurons in the telencephalic wall, the SP neurons also form the first synapses in the telencephalic wall. Therefore, the SP neurons are the first neurons in the telencephalon capable of performing adult-like functions and important for the proper functional organization of the future cerebral cortex.

Initial autoradiographic studies of SP neurons in rhesus monkeys, cats, and rodents clearly demonstrated that WMIN are remnants of the fetal SP neurons [6,9,10,11,12,13]. Many subsequent studies confirmed and extended these findings and described various morphological and molecular phenotypes of both SP neurons and WMIN [6,13,14,15,16,17,18,19]. It was already known that WMIN are remnants of SP neurons, so the next logical step was to investigate how many SP neurons survive as WMIN in the adult brain. Initial studies in cats and rodents suggested that up to 80% of the fetal SP neurons undergo apoptosis during the perinatal or early postnatal period [7,10,11,12,15]. However, such widespread apoptosis of SP neurons was never observed in the human or monkey brain [6,9,20], suggesting that there may be inter-species differences in the developmental fate of WMIN. Moreover, subsequent experimental and quantitative studies demonstrated that the apoptosis of SP neurons is much less pronounced even in rodent and cat brains [21,22].

To understand the morphological and molecular heterogeneity of WMIN in the adult brain, it is important to realize that the population of fetal SP neurons is composed of various neuronal types, similar to the population of cortical neurons. The SP neurons are derived from several neurogenetic sources. While the neurons which initially form the subplate zone are among the earliest generated neurons in the telencephalon [6,7,9,10,23], additional neurons are continually added as the subplate develops further. Thus, different SP neurons have different developmental origins and molecular profiles [19,24,25,26]. This later addition of new neurons is quite pronounced and long-lasting in the developing human brain [6,27,28]. This finding opens the possibility that certain subsets of SP/WMIN neurons are significantly (or even exclusively) expanded in the human brain in comparison to the brains of experimental animals [28].

As already mentioned, not all subsets of SP neurons in experimental animals undergo the same amount of apoptosis. For example, in mice, the late-generated SP neurons which express *Nurr1, Lpar1*, and *Cplx3* selectively survive apoptosis [19]. While similar studies were not conducted on human brain tissue, it stands to reason that similar selective survival of SP neurons would also be present in the human brain. However, as the relative number of WMIN in humans is in general significantly higher than in rodents, it is also possible that the late-generated human SP neurons survive in disproportionally large numbers.

In all species analyzed to date, WMIN are present throughout the entire subcortical white matter [3,6,9,13,15,16,17,19,21,29,30,31,32]. They display morphological and molecular heterogeneity almost equal to that of cortical neurons in the remaining six cortical layers and include both pyramidal and non-pyramidal types [9,16,17,18,33] (Figure 1). The exact proportion of each morphological type remains to be determined. Most studies reported WMIN as predominantly fusiform and polymorph [9,18,33], and a few studies suggested that WMIN are predominantly pyramidal [16]. WMIN express markers of both excitatory and inhibitory neuronal populations. All WMIN are positive for the pan-neuronal marker NeuN [17,32,34]. They express several markers of the glutamatergic phenotype such as vGLUT1, MAP2, and SMI32 [16,17,35]. WMIN also express a large number of GABAergic markers such as the calcium-binding proteins calbindin, calretinin, parvalbumin, the GABA transporters GAT1, vGAT [17] (Figure 2), the peptidergic neurotransmitters NPY [36,37,38,39,40], cholecystokinin [41], avian pancreatic polypeptide [33], somatostatin [38,41,42], substance P [41,43], as well as markers of other neurotransmitter systems: NADPH/NOS [16,17,18,38,44,45,46,47,48], AChE [9,20,47]. It is interesting to note that the overwhelming majority of the cortical NOS/NADPH neurons (over 80%) are located in the white matter [18,38].

To date, only two studies have analyzed the synaptic distribution on WMIN in the human brain [9,17]. Both studies found symmetrical and asymmetrical synapses located on the soma and dendrites of WMIN but described a different distribution of these synapses; in fact, according to Kostović and Rakic [9], axosomatic synapses are symmetrical and asymmetrical and axodendritic are asymmetrical, whereas, according to Garcia-Marin et al. [17], axosomatic are symmetrical and axodendritic are both symmetrical and asymmetrical. Both studies found that the density of synapses on WMIN is low and decreases with the depth of the white matter [9,17]. It should be noted that the existence of synapses does not prove that these synapses are functional (i.e., synapses can be “silent”), though it demonstrates that neurons are involved in forming neural circuits (even if these circuits remain “silent”). During development, SP neurons receive axons from both subcortical and cortical sources, and these axons establish temporary synaptic connections with SP neurons; during later development, the majority of these afferent axons relocate from the SP into the cortical plate (for review, see [23]). It would be important to know if any of these transient fetal connections (of SP neurons) are retained in surviving WMIN in the adult brain. A pioneering study of that issue [50] demonstrated that, in cats, at least some synapses on WMIN originate from axons of overlying cortical neurons. It is also known that WMIN axons project to the overlying cortical layers, including the cortical layer I [51,52].

## 3. Total Neuronal Number, Density, and Spatial Distribution of WMIN

In order to fully understand the importance of WMIN in the brain, one must consider the size and spatial distribution of this neuronal population. The quantitative studies of the number of WMIN have been significantly influenced by three factors: (a) the assumption that the majority of subplate neurons undergo apoptosis; (b) the assumption that the surviving WMIN represent a pathological finding; and (c) existing problems with a proper delineation of WMIN-containing compartments. These issues have contributed to the persistence of the notion that WMIN are a rather small neuronal population in the human brain, probably significant only in pathological cases. Therefore, an important message of this review is that WMIN in the human brain in reality represent a significant (at least 3%) subset of all cortical neurons and thus probably have certain important (but still undefined) functional roles.

It should be pointed out that there is a bewildering variety of approaches to counting WMIN. The main reason for such diversity is the underlying diversity in conceptual approaches of how to define the region of interest when counting WMIN, as well as how to correctly define the WMIN population. In addition to that, many studies failed to apply proper stereological criteria. At present, the generally accepted definition of WMIN is that they are a subset of neurons located in the white matter below the cerebral cortex. This immediately raises two important questions: (a) where exactly is the border between the deepest cortical layer and the white matter? and (b) how deep into the white matter do WMIN “normally” extend? It should be noted that, in the adult human brain, there are two distinct populations of WMIN [18]: (a) deep WMIN located in the periventricular white matter (in the vicinity of basal ganglia) and (b) superficial WMIN, that is WMIN located in the gyral/sulcal white matter immediately below (up to 3 mm) the cerebral cortex—that is, in the white matter segment IV of the classical division [53], which corresponds to the fetal SP compartment and perinatal/early postnatal subplate remnant [18,23]. The superficial WMIN population is the one usually referred to as WMIN in most current publications. It should be noted that the distinction between deep and superficial WMIN populations is easily noted in the adult human (or monkey) brain, because they are separated by the wide von Monakow’s segment III of the white matter (centrum semiovale). However, in small experimental rodents, these populations are close to each other and thus difficult to separate. Our survey of the existing literature demonstrated that in most studies there was not a uniform delineation of the WMIN compartment and the region of interest for the purpose of counting WMIN was usually arbitrarily defined. The approaches ranged from hand-marking a part of white matter which would contain WMIN [54,55] to placing pre-defined boxes within the white matter to count WMIN [56] and defining the width of white matter below the cerebral cortex which would be considered as the WMIN compartment [17,32,57,58]. Obviously, these inconsistencies can significantly influence the estimation of the WMIN population (Figure 3). At present, when counting brain neurons, one should apply stereology approaches as a gold standard. However, in using stereology, one has to follow several strict rules—such as to precisely determine the region of interest. The satisfaction of this criterion is prominently lacking in most published studies on the number of WMIN. The part of the white matter which should represent the “WMIN compartment” has been defined in significantly different ways, thus leading to significantly different WMIN counts (for details, see [32]).

WMIN density is not uniform across the cerebral cortex, varying between different areas and different parts of the gyrus. The highest number of WMIN can be observed at the gyral crown, and the lowest number at the bottom of the sulcus [17,32]. Furthermore, the density of WMIN decreases with depth from the cortex/white matter border [17,32,54]. Using pan-neuronal WMIN markers, the reported density of WMIN in the human brain ranges from 1000 neurons/mm^3^ to 3000 neurons/mm^3^ [17,32,54,55,59,60]. There is no consensus on the region with the largest density of WMIN, with some studies reporting the highest density in the frontal cortex [17,32], and others reporting it in the temporal cortex [54]. Similarly, the area with the lowest density of WMIN has been variously reported as the cingulate cortex [32], the temporal cortex [17], or the frontal cortex [54]. The observed differences can be explained by different definitions of the WMIN compartment and sampling protocols. The density of WMIN can significantly vary based on the size of the WMIN compartment (as they are denser when closer to the white matter/cortex border), their location within the gyrus (denser in the gyral crown), and the type of gyrus (denser in the smaller gyrus than in the larger gyrus); all these approaches can artificially increase or decrease the density of WMIN [32]. Neuronal density, although instructive with respect to neuronal position, still does not provide us with information about the size of the WMIN population. As pointed out above, WMIN density greatly varies between areas and with subcortical depth. Several recent studies attempted to count the total number of WMIN using a stereological approach. In these studies, the total number of WMIN ranged from 600,000,000 to 1,100,000,000, indicating that WMIN represent a large neuronal subpopulation in the human brain [32,57]. The large range of the total number of WMIN could be explained by significant inter-individual differences or differences in the WMIN compartment used for counting [32,57]. Without a proper definition of the WMIN compartment, it is impossible to compare data from the different studies. In our recent publication [32], we proposed that the von Monakow segment IV should be considered as the WMIN compartment, as per definition, WMIN are located in the gyral white matter, which is the von Monakow segment IV; the adult segment IV is composed of short cortico–cortical fibers which, during development, invade the upper part of the transient SP zone. Therefore, as WMIN are remnants of the SP neurons, we should use the von Monakow segment IV of white matter as the adult proxy of the fetal SP zone. Although WMIN density and the total number of WMIN vary greatly between studies, one cannot neglect the fact that WMIN are a large and significant subset of cortical neurons. If we compare data about the number of WMIN with those published about the total neuronal number in the cerebral cortex and some other important brain structures [61,62], we can see that WMIN are more numerous than neurons in the globus pallidus (400×), amygdalae (50×), claustrum (40×), entorhinal cortex (40×), Purkinje cells (30×), thalamus (10×), caudate nucleus (10×), putamen (8×), etc. The total number of neurons in the human cerebral cortex ranges from 10 to 20 billion [62,63], and when these numbers are compared with the total number of WMIN (0.6 to 1.1 billion), we can conclude that the WMIN represent a significant neuronal population. Estimates range from as low as 3% of all cortical neurons (0.6 billion WMIN in 20 billion cortical neurons) to 10% (1.1 billion WMIN in 10 billion cortical neurons). In our opinion, WMIN could represent around 5% of neurons in the human cerebral cortex.

The WMIN compartment is not populated only by neurons. The majority of cells located in the compartment belong to the glial lineage. All three major glial classes (astrocytes, oligodendrocytes, and microglia) can be found in the WMIN compartment. The exact number and composition of glial cells in the WMIN compartment needs to be studied in the future. However, it is prudent to conclude that glial cells’ composition would be similar to that observed in the other parts of the white matter. In the white matter, the reported glial density ranges from 20,000 to 200,000 cells per mm^3^ [63]. The most frequent type of glial cells are oligodendrocytes (45–75%), followed by astrocytes (19–40%) and microglia (around 10%) [63]. In recent years, a significant amount of data was collected indicating that glial cells are not passive elements of the white matter but are actively involved in synaptic development and plasticity and in the regulation of neuronal activity through tripartite synapses [64,65,66,67]. Glial cells could significantly impact the information flow in the cortical circuitry by influencing WMIN (see WMIN function below). It is interesting to note that many adverse events during the late fetal and perinatal periods (such as ischemia or hemorrhage) occur in the subplate zone at a time critical for the generation of oligodendrocytes and astrocytes [23,68]. Therefore, disruption of glial migration could lead to the observed disturbance in myelination but could also significantly influence the future signal processing of WMIN by disrupting the normal organization and composition of glial cells in the WMIN compartment.

## 4. Functional Importance of WMIN

Although WMIN were discovered over 150 years ago, there are still very few data about their functional importance. Only a handful of studies investigated the functional properties of WMIN. Several studies examined the electrophysiological properties of WMIN in both rodent and human brain. The findings of these studies indicate that WMIN are an active, fully functional group of neurons integrated in the cerebral circuitry [25,30,50,69,70,71,72]. WMIN and SP neurons have similar electrophysiological properties; however, WMIN also exhibit some specific electrophysiological features indicating that they develop and mature further during the postnatal period [71]. Furthermore, these studies indicated that although WMIN are located close to the cortical border, their functional properties are different from those of adjacent cortical neurons. For example, WMIN neurons showed a lower depolarization threshold and a different response to a stimulus in comparison to cortical neurons. Based on electrophysiological experiments, WMIN receive both a glutamatergic and a GABAergic input [71].

Only a few studies discussed potential functional roles of WMIN, and mostly offered some speculative suggestions based on morphological, spatial, and molecular data. One of the functions attributed to WMIN is the regulation of the cerebral cortical blood flow [37,73,74]. A large subset of WMIN consist of nitrinergic neurons, and their axons can be observed apposed to blood vessels [18,73,74]. As nitric oxide is one of the most potent vasodilators, this observation gave rise to the idea that WMIN link the brain function with blood flow, i.e., increasing the blood flow when there is an increased brain activity and decreasing the blood flow when there is a suppressed brain activity in a small part of the cerebral cortex [73,74,75,76]. WMIN have also been implicated in the process of sleep regulation [77]. Another proposed function of WMIN is the control of information flow to the cerebral cortex [1,30,78,79]. WMIN are positioned at a critical location (cortical–white matter interface) in the brain, where they can significantly influence the information flow to the cortex. Furthermore, WMIN axons and dendrites project heavily to the adjacent cerebral cortex, and some WMIN send their axons even to the cortical layer I [15,30,50]. Taken together, all these data indicate that WMIN are active participants in the cortical circuitry, influencing data processing within the cerebral cortex.

## 5. Pathology of WMIN

A number of studies have proposed that WMIN may be involved in various brain disorders, such as epilepsy [55,59,80,81,82,83,84], schizophrenia [34,35,44,45,60,85,86,87,88,89,90,91,92,93], depression [88,94], bipolar disorder [88,91], autism spectrum disorder [95], Alzheimer’s disease [41,43,96,97,98], multiple system atrophy [57], etc. Epilepsy is one of the disorders most often associated with a putative WMIN pathology [55,59]. The reason for this is the observation that in some patients suffering from epilepsy, an increased density of neurons in the white matter has been observed [55,59,81,82,83,84]. This observation initially served as evidence of WMIN being a pathological finding in the human brain [4]. Although most researchers today believe that an increased density of neurons in the white matter is the result of arrested migration of cortical neurons in the white matter [55,59,81,82,83,84], there is no direct evidence for this hypothesis. One of the reasons is the lack of specific WMIN markers which could differentiate true WMIN from other cortical neurons arrested in their migration. As we have shown in previous sections, WMIN and cortical neurons share many biomarkers and are virtually indistinguishable by these markers alone. The discovery of WMIN-specific markers would greatly improve our understanding of WMIN biology and resolve the dilemma about extra neurons observed in the white matter of many epileptic cases.

The strongest involvement of WMIN in pathology can be observed in schizophrenia. Many studies showed alterations in WMIN density, spatial distribution, neuronal composition, molecular expression, and synaptic properties [34,35,44,45,47,60,86,87,88,89,90,99]. In studies analyzing WMIN alterations in schizophrenia, the main findings were changes in WMIN density and spatial distribution, e.g., increased density of NeuN- and MAP2-positive WMIN [34,35,60,86,87] and decreased density of NADPH-positive WMIN [44,45]. Another important observation is the change in the spatial distribution of, for example, NADPH-positive WMIN which display an increased density close to the cortex–white matter border [44,99]. However, the results of these studies are somewhat inconsistent. While most disturbances were reported in the superficial white matter, some studies (even from the same authors) reported no changes in the superficial white matter and, instead, reported changes in the deep white matter. It is important to note that both superficial and deep white matter are a part of the gyral white matter (von Monakow segment IV) and that the division between them is arbitrary, often done differently in different studies. Therefore, one of the reasons for the observed inconsistencies could be the artificial division of the gyral white matter. It is interesting to note that the observed disorders were not present in all types of schizophrenic patients. The biggest changes were observed in patients with negative symptoms [34,86,87]. These findings further suggest that schizophrenia is a spectrum disorder rather than a single disorder and that, in at least one subset of patients, WMIN may play an important role in the pathogenesis of the disorder. Another disorder with WMIN involvement is Alzheimer’s disease, where the observed pathologies of WMIN are similar to those observed in cortical neurons, but primarily concern the subset of WMIN expressing somatostatin [96,98].

## 6. Conclusions

Although WMIN were described over 150 years ago, they are frequently considered a small and relatively unimportant neuronal population. Since their identification, a significant amount of data on WMIN were collected. WMIN comprise many different morphological types, may exhibit both glutamatergic and GABAergic phenotypes, many peptidergic neurotransmitters, and a lot of structural proteins that can be observed in cortical neurons. As most of the data were obtained from studies which primarily focused on cortical neurons, WMIN never received a proper thorough scrutiny. As a result, WMIN are still lacking a proper definition (based on their location, morphology, physiology, and phenotype) to reliably recognize them. Often, the definition of WMIN is circumstantial, based on their spatial location, i.e., within the white matter. Recent attempts to characterize and standardize the WMIN compartment using developmental criteria, similarly to the way cortical neurons are defined, allowed to perform more accurate studies on WMIN. These studies used a stereological approach and demonstrated that WMIN are a significant subpopulation of cortical neurons. However, the issue about differentiating cortical neurons from WMIN at the border of cortex and white matter, which impacts the size of the WMIN population and its importance, remains. Currently, there are several approaches to address this problem. One of the approaches is to develop an automatic computational algorithm for delineating cortical layers. Preliminary results look promising, and if successful these automatic delineation algorithms could efficiently and reliably determine which neurons belong to the WMIN population and which to the cortical population. Another approach is to find a WMIN-specific biomarker that would help define this subpopulation. The quest for a unique WMIN biomarker has been exceedingly difficult, as WMIN and cortical neurons are generated from the same proliferative zone and share many biomarkers. The recent use of transgenic animals has provided us with some potential candidates for a WMIN-specific biomarker. The elucidation of a specific WMIN biomarker would not only significantly help determine the borders of the WMIN population, especially the white matter–cortex border, but also help future studies on the molecular profile and physiological properties of WMIN. Although high-throughput transcriptomics and proteomic studies are common in neuroscience, WMIN have not been extensively scrutinized by these techniques. One of the major reasons is the difficulty in differentiating between cortical neurons and WMIN. Therefore, the successful elucidation of a WMIN biomarker would provide us with a chance to successfully apply these novel techniques and enhance our understanding of the normal physiology and phenotype of the WMIN population. A better understanding of their normal role and involvement in cortical circuits will enable the elucidation of their putative role in several brain disorders and thus open new avenues in combating these pathologies. While WMIN studies in experimental animal models should definitely continue, it is important to keep in mind the significant species-specific differences in the number, distribution, and putative functions of WMIN. Therefore, the study of WMIN in the human brain should involve a significant part of our future efforts.

## Figures and Tables

**Figure 1 cells-10-00190-f001:**
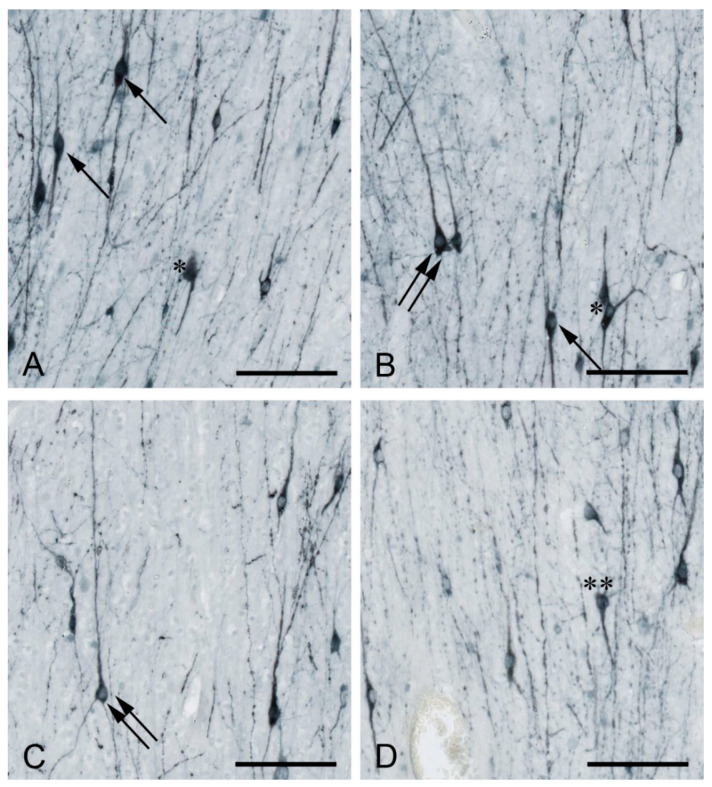
MAP2-positive white matter interstitial neurons (WMIN) in various parts of the human frontal cortex. WMIN comprise various morphological types such as bipolar (arrow in (**A**,**B**), pyramidal (double arrow in (**B**,**C**)), triangular/multipolar (asterisk in (**A**,**B**)), and “inverted pyramidal” (double asterisk in (**D**)) neurons. Bar = 100 µm. Samples are part of the Zagreb Neuroembryological Collection [49].

**Figure 2 cells-10-00190-f002:**
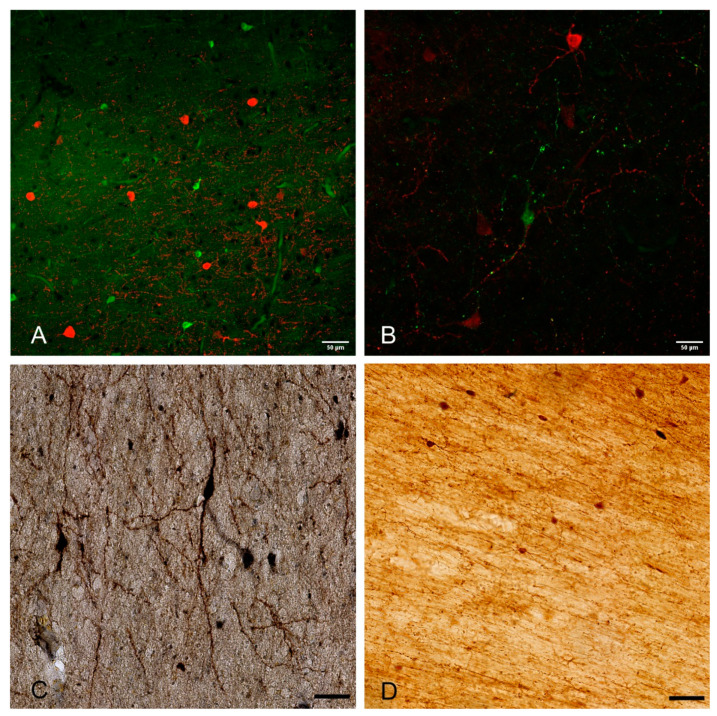
Examples of WMIN in the human brain (**A**–**C**) and rhesus monkey brain (**D**). The depicted WMIN are positive for calretinin (green in (**A**,**B**,**D**)), parvalbumin (red in (**A**)), calbindin (red in (**B**)), and nNOS (**C**)). Note that all neurons in both human and rhesus monkey show extensive dendritic arborization. Bar = 50 µm. We would like to thank Professor Zdravko Petanjek and Professor Monique Esclapez for the images.

**Figure 3 cells-10-00190-f003:**
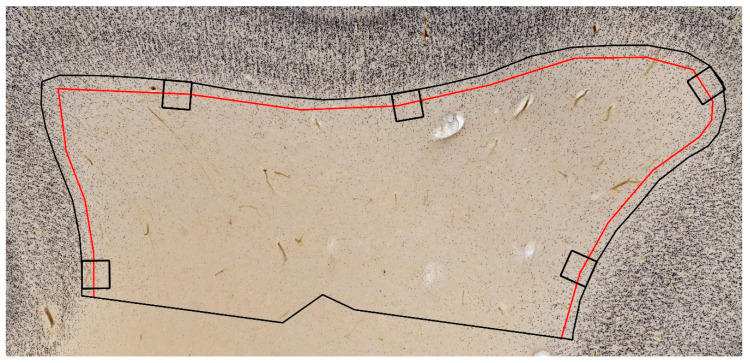
Illustration of different regions of interest for WMIN counting. Note the significant differences in surveying WMIN using pre-defined squares (black squares), an arbitrary region under the cortex (here 175 µm, red line), and the prospective von Monakow segment IV (defined here as 3 mm below cortex, black line). Note that the first two approaches leave out many of the WMIN, especially in the deeper parts of the white matter. The image is part of Zagreb Neuroembryological Collection [49].

## Data Availability

No new data were created or analyzed in this study. Data sharing is not applicable to this article.

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
