# Peer review of "White Matter Interstitial Neurons in the Adult Human Brain: 3% of Cortical Neurons in Quest for Recognition"

_cells, 2021, doi:10.3390/cells10010190_

Round 1

Reviewer 1 Report

-SP in the abstract has never been explained before

-the abstract in its final part seems inconclusive because it does not express the purpose of the study

-on page 5, 3% do not appear to be supported by data. The author should explain analytically how he arrives at this conclusion in a separate paragraph

-the author should highlight and discuss in a separate paragraph, the data of his study distinct from those of the current literature

-the author should explain how his finding impacts the current state of the art

Author Response

  • SP in the abstract has never been explained before

We thank the reviewer for mentioning this error. We have spelled the abbreviation in the abstract and checked the entire manuscript for additional abbreviations.

  • The abstract in its final part seems inconclusive because it does not express the purpose of the study

We added the sentence at the end of the abstract stating that this is the review article. The sentence is: “In this review we will provide an overview of current WMIN literature with an emphasis on studies conducted on the human brain.”

  • On page 5, 3% do not appear to be supported by the data. The author should explain analytically how he arrives at this conclusion in a separate paragraph

As this is the review article, we did not present any novel data, but based our estimate from the available literature on the total number of WMIN and cortical neurons. The 3% is the low estimate of the WMIN percentage in the total cortical population. However we added a new reference (number 63 in the text) and a short paragraph, on page 7 about how we estimated 3%.

The paragraph reads: “The total number of neurons in the human cerebral cortex ranges from 10 to 20 billion neurons [62, 63] and when these numbers are compared with the total number of WMIN (0.6 to 1.1. billion) we can conclude that the WMIN represent a significant neuronal population. Estimates range from as low as 3% of all cortical neurons (0.6 billion WMIN in 20 billion cortical neurons) up to 10% (1.1 billion WMIN in 10 billion cortical neurons). In our opinion, WMIN could represent around 5% of neurons in the human cerebral cortex.

  • The author should highlight and discuss in a separate paragraph, the data of his study distinct from those of the current literature

We did not present any new data in this manuscript, as this is a review article.

  • The author should explain how his finding impacts the current state of the art.

As this is not an original study, but a review of current literature there are no original findings that can be assessed for the impact on the current state of the art. In the manuscript we reviewed the state of the art in the field or lack of it (one of the motivation for writing this review was a lack of use of modern techniques, such as transcriptomics, on WMIN).

Reviewer 2 Report

The review by Sedmak and Judas is an important review that addresses a gap in literature on White matter interstitial neurons. While the manuscript is scientifically sound, there are few points that should be addressed before becoming suitable for publication:

1- A short section about different type of neurones is required. Some neutrons (i.e. SP neurons) are discussed with our even being introduced by abbreviation. 

2- For the provided images, experimental data are missing. I realize this is a review article, and not an original paper, but the work (if unpublished already) should include all information about the sample collection, experimental procedures, material & methods, etc. If already published, the source should be identified (a permission to use, if applicable).

3- A careful proofread of the format, style, and grammar should be done.

4- All abbreviations should be mentioned in the first usage.

5- Although the manuscript is about neutrons, a few sentences on glial cellular compartments of the white matter would improve the manuscript.

Author Response

  1. A short section about different type of neurones is required. Some neutrons (i.e. SP neurons) are discussed with our even being introduced by abbreviation.

We added a small part about the composition of SP and SP neurons on page 2.

The paragraph reads: “The SP is composed of mature postmitotic neurons, afferent and efferent axons, and abundant extracellular matrix. The SP neurons exhibit both glutamatergic and GABAergic profiles. It is important to note that, besides being among the earliest generated neurons in the telencephalic wall, the SP neurons also form the first synapses in the telencephalic wall. Therefore, SP neurons are the first neurons in the telencephalon capable of performing adult-like functions and important for proper functional organization of the future cerebral cortex.”

  1. For the provided images, experimental data are missing. I realize this is a review article, and not an original paper, but the work (if unpublished already) should include all information about the sample collection, experimental procedures, material & methods, etc. If already published, the source should be identified (a permission to use, if applicable).

As the reviewer mentioned, this is a review article and not an original contribution. The images presented here are not novel findings, but common knowledge and experiments were not performed exclusively for this publication. The images were obtained from the archival tissue sections which are part of the Zagreb Neuroembryological Collection, or graciously provided by our collaborators form their archival material. We felt that it is better to add our archival images, for which we do not need permission, than to use already published images from other journals. The images were added to demonstrate few points about WMIN to the readers without the need to consult the cited literature. Therefore, we believe that adding the material and methods section to the manuscript would confuse the readers about true nature of the manuscript (as material and methods are not included in the review articles except in special cases). If reviewer wishes we can remove the images from the manuscript, as these images are for the purpose of illustration and not corroboration of findings.

  1. A careful proofread of the format, style, and grammar should be done.

The manuscript has been proofread by a professional.

  1. All abbreviations should be mentioned in the first usage.

We have checked the manuscript for the abbreviations and corrected where they were not mentioned.

  1. Although the manuscript is about neutrons, a few sentences on glial cellular compartments of the white matter would improve the manuscript.

We added a small paragraph about glial component on page 7 and possible role of glia in WMIN function. The paragraph reads: “The WMIN compartment is not populated only by neurons. The majority of cells located in the compartment belong to the glial lineage. All three major glial classes (astrocytes, oligodendrocytes and microglia) can be found in the WMIN compartment. The exact number and composition of glial cells in the WMIN compartment needs to be studied in the future. However, it is prudent to conclude that the composition would be similar as observed in the other parts of the white matter. In the white matter, the reported glial density ranges from 20.000 to 200.000 cells per 1 mm3 [63]. The most frequent type of glial cells are oligodendrocytes (45 – 75%), followed by astrocytes (19 – 40%) and microglia (around 10%) [63]. In recent years, a significant amount of data was collected indicating that glial cells are not only passive elements of the white matter, but are actively involved in the synaptic development, plasticity and the regulation of neuronal activity through tripartite synapses [64-67]. Glial cells could significantly impact the cortical circuitry information flow by influencing WMIN (see WMIN function below). It is interesting to note that many adverse events during a late fetal and perinatal period (such as ischemia or hemorrhage) occur in the subplate zone during the time period critical for the generation of oligodendrocytes and astrocytes [23, 68]. Therefore, the disruption of glial migration could lead to the observed disturbance in myelination, but could also significantly influence the future signal processing of WMIN by disrupting a normal organization and composition of glial cells in the WMIN compartment.

Round 2

Reviewer 1 Report

The present form is not still adequate to the aims and scope of the title.

The advices are the same 

Author Response

We thank the reviewer for the comments on our revised manuscript. However, we respectfully disagree with the reviewer’s comments. From the previous comments we are unable to conclude which “data from this study” we should discuss. As we already mentioned there is no new data presented in this review. All data have already been discussed in their respective publications, and here we report these data as presented in the original papers. Furthermore, as there is no new data we cannot elaborate on the “impact of findings on the field”. The only data produced in this manuscript is the notion that WMIN are poorly understood, often overlooked neuronal population. Therefore, we stand by our previous response to the specific issues.

However, we reviewed our manuscript and decided to re-write the Conclusion part. In the conclusion part we added a part about future direction of research and current attempts to tackle some of the issues plaguing the WMIN research. Hopefully, these additions will be helpful for the readers. The new conclusion part now reads:

Although, the WMIN were described over 150 years ago, they are frequently considered as a small and relatively unimportant neuronal population. During this period, a significant amount of data on WMIN were collected. The WMIN exhibit many different morphological types, both glutamatergic and GABAergic phenotype, many peptidergic neurotransmitters, and a lot of structural proteins that can be observed in the cortical neurons. As most of the data were obtained from studies which primarily focused on the cortical neurons, the WMIN never received proper thorough scrutiny. As a result, the WMIN are still lacking the proper definition (based on the location, morphology, physiology, and phenotype) to reliably recognize them. Often, the definition of WMIN population is circumstantial based on the spatial location, i.e. located in the white matter. Recent attempts to characterize and standardize the WMIN compartment using the developmental criteria, similarly as the cortical neurons are defined, allowed to perform more accurate studies on the WMIN. These attempts resulted with the several studies, using stereological approach demonstrating that WMIN are a significant subpopulation of cortical neurons. However, there is still an issue with differentiating cortical neurons and WMIN on the border of cortex and white matter which impacts the size of the WMIN population and its importance. Currently, there are several approaches to address this problem. One of the approaches is to develop an automatic computational algorithm for delineating cortical layers using an automatic approach. The preliminary results look promising, and if successful these automatic delineation algorithms could efficiently and reliably determine which neurons belong to the WMIN population and which to the cortical population. The other approach is to find a WMIN specific biomarker that would help define the subpopulation. The quest for the unique WMIN biomarker has been exceedingly difficult, as the WMIN and cortical neurons are generated from the same proliferative zone and share many biomarkers. Recent use of transgenic animals provided us with some potential candidates for the WMIN specific biomarker. The elucidation of the specific WMIN biomarker would not only significantly help in determining the borders of the WMIN population, especially at the white matter – cortex border, but also help future studies on the molecular profile and physiological properties of the WMIN. Although the high-throughput transcriptomics and proteomic studies are common in the neuroscience, the WMIN have not been extensively scrutinized by these techniques. One of the major reasons is the difficulty in differentiating between cortical neurons and the WMIN. Therefore, successful elucidation of the WMIN biomarker would provide us with the chance to successfully apply these novel techniques and enhance our understanding of the normal physiology and phenotype of the WMIN population. Better understanding of their normal role and involvement in cortical circuits will enable the elucidation of their putative role in several brain disorders, and thus open new avenues in combating these disorders. While WMIN studies in experimental animal models should definitely continue, it is important to keep in mind the significant species-specific differences in the number, distribution and putative functions of WMIN. Therefore, the study of WMIN in the human brain should represent a significant part of our future efforts.

Reviewer 2 Report

No further comments.

Author Response

We thank the reviewer for the time and effort in revising our manuscript.